# Economic Feasibility of Retrofitting an Ageing Ship to Improve the Environmental Footprint

Dimitar Yalamov [1], Petar Georgiev [1] and Yordan Garbatov [2,*]

1. Naval Architecture and Marine Engineering Department, Technical University of Varna, 1, Studentska Str., 9010 Varna, Bulgaria
2. Centre for Marine Technology and Ocean Engineering (CENTEC), Instituto Superior Técnico, Universidade de Lisboa, 1049-001 Lisbon, Portugal
* Correspondence: yordan.garbatov@tecnico.ulisboa.pt

**Abstract:** Natural gas is cheaper than fuel on an energy basis, making it an alternative ship fuel which leads to a reduced operating cost and clean gas environmental conditions. The current study analyses the retrofit of an ageing multi-purpose ship to use liquefied natural gas as a primary ship fuel in the context of a short-ship sea operation. The objective is to transform an existing commercial ageing ship propulsion system into a green energy propulsion one and to analyse the economic feasibility considering the high volatility and increased LNG price. Four scenarios were analysed based on the net present value representing Denying, Disinterested, Good and Acceptable financial cash outflow. It was concluded that in the present economic instability and price of LNG fuel and $CO_2$ taxes, the ship owner needs to rely on the long-term contract of buying LNG fuel to implement measures to reduce greenhouse gas emissions and keep good benefits in shipping.

**Keywords:** retrofitting; ship; ageing; decarbonisation; short sea shipping; green energy; net present value





## 1. Introduction

The recent estimates of the Fourth International Maritime Organisation (IMO) Greenhouse Gas (GHG) Study 2020 [1] show a troubling message for shipping Greenhouse Gas emissions. There was a 9.6% increase in GHG emissions from 2012 to 2018, mainly due to a continuous increase in global maritime trade. The share of shipping emissions also increased from 2.76% in 2012 to 2.89% in 2018.

The activities of IMO addressing climate change started more than ten years ago by signing a Cooperation Agreement between IMO and the Korea International Cooperation Agency (KOICA) on April 2011 for the implementation of a technical cooperation project on Building Capacities in East Asian countries to address Greenhouse Gas Emissions (GHG) from Ships [2]. In recent years, numerous resolutions and documents have been adopted [3] to achieve the ambitious goal of a 50% reduction in the total annual GHG emissions and 70% reduction in $CO_2$ emissions per transport work compared to 2008 by 2050. For the implementation of these goals, several mandatory instruments for new and existing ships have been proposed, such as the MARPOL Annex VI, with the Energy Efficiency Design Index (EEDI) introduced increasingly strict carbon intensity standards for new ships and the Ship Energy Efficiency Management Plan (SEEMP) for operators to improve the energy efficiency of all ships and, additionally, the Energy Efficiency Operational Indicator (EEOI), Energy Efficiency Existing Ship Index (EEXI) and Carbon Intensity Indicator (CII).

Generalisations and guidelines for increasing the energy efficiency of ships have appeared after the first approved requirements [4]. They were in several groups: Hull form optimisation; Energy-saving devices; Structural optimisation and light-weight construction; Machinery technology; and Fuel efficiency of ships in service. Several examples of structural optimisation and light-weight construction can be seen in [5–9] and for control of ship operation due to the generated air pollution by ships in coastal water in [10–13].

The development of machine technologies is also related to the introduction of alternative fuels. Even in the early years, special attention was paid to Liquified Natural Gas (LNG) [14–16]. The LNG is cleaner than coal or oil and its use as fuel in shipping leads to a reduction in $NO_x$, $SO_x$ and $CO_2$ emissions. This is of great importance for operations in the Emission Control Area (ECA), where 18.5% of handy-size tankers and small and medium size Ro-Ro spend 80% of their sailing time [15].

The use of LNG complies with ECA regulations as sulphur oxides emissions are reduced to zero. The nitrogen-oxide emissions are below the Tier III regulation, applicable in ECA from 2016, and it is very low in particles reducing at the same time the carbon dioxide emissions in the range of 20–25% [16].

A comparison of LNG's life cycle environmental performance, liquefied biogas (LBG), methanol and bio-methanol are presented in [17], where four aspects are considered, i.e., technical, economic, environmental and other. The last group are safety and safe handling, availability, public opinion, etc. Although liquefied natural gas or methanol produced from natural gas significantly improves environmental performance, the impact on the climate is of the same order as using heavy fuel. There will be a more significant effect when using methane and methanol produced from biomass.

Alternative fuels, in combination with additional equipment, are an option to meet the requirements of ECA [18]. A comparison of three alternatives: Heavy Fuel Oil (HFO) combined with a scrubber and a Selective Catalytic Reduction (SCR), Marine Gasol (MGO) combined with an SCR and LNG shows that none of them leads to less impact on the climate than heavy fuel. Reducing the methane slip to 2 wt.% (weight percentage) would ensure that LNG has a lower impact on climate change.

Comparison employing the Life Cycle Assessment (LCA) of specific ships operating in a particular area may show an advantage when using one or another fuel. This way, a feeder container ship and a passenger ferry operating between Mainland China and Taiwan are compared [19]. For both ships, two scenarios are considered with HFO and LNG as fuel. The study indicated possible improvement in total fuel-life cycle GHG emissions from using LNG. The reduction in the emissions of $NO_x$ is (38–39%) and CO (42–43%). Remarkable reduction is obtained using LNG in $SO_2$ (99.8%) and $PM_{10}$ (97.5%), while methane emissions increase significantly when LNG is used as an alternative fuel for both ships.

Comparative studies also appear with the continuous improvement of technologies to produce alternative fuels. A recently presented study [20] compares seven alternative marine fuels—LNG, liquefied biogas (LBG), methanol from natural gas, renewable methanol, hydrogen for fuel cells produced from (1) natural gas or (2) electrolysis based on renewable electricity, and Hydrotreated Vegetable Oil (HVO), and HFO. Ten performance criteria rank the marine fuels and different stakeholder groups set their relative importance in Sweden. The criteria include economic, environmental, technical and social aspects. Economic criteria are the most important for ship owners, fuel producers and engine manufacturers, while the Swedish government authorities prioritise environmental criteria, specifically GHG emissions. Based on the views of the first group members, LNG is ranked the highest, with HFO second, then the fossil methanol, followed by biofuels. For the second group of stakeholders, hydrogen is ranked highest, followed by renewable methanol and HVO.

The Getting to Zero Coalition [21] is an alliance of more than 200 companies and organisations from the maritime, energy, infrastructure and financial sectors, supported by governments and international organisations. The goal of the coalition is to bring into operation commercial alternative fuels with zero emissions by 2030, leading to the complete decarbonisation of maritime transport by 2050. To achieve this goal, an S-curve has been defined [22] according to which, by 2030, scalable zero-emission fuels (SZEF) as hydrogen and hydrogen-derived fuels, such as ammonia, e-methanol and synthetic hydrocarbon, should be 5% of all fuels used, 27% of all fuel used by 2036 and 93% of all fuel used by 2046. Such a pace is needed to meet the goal of the 2016 Paris climate agreement to limit global warming to well below 2, preferably to 1.5 degrees Celsius, compared to pre-industrial

levels [23]. A major obstacle to achieving these goals is the significant price gap between fossil and zero-emission fuels. For example, the estimated production price (EUR/GJ) by 2025 of e-hydrogen, e-ammonia, e-methane and e-methanol is 6 to 7.3 times higher than that of LNG [24].

A summary of the use of alternative fuels as of 2022 was made by Det Norske Veritas [25]. Of the total gross tonnage of ships operating today, 5.5% use alternative fuels and one-third (33%) are on order and can work with alternative fuels. The report evaluates 24 scenarios for the maritime energy mix in 2050 among five groups: fossil fuels, biofuels, electro fuels, blue fuels and electricity. A preferred alternative fuel cannot be specified due to uncertainty about the availability of sufficient amounts of biomass for biofuels or sufficient renewable electricity for electric fuel production. In addition, considerable investments in this direction are needed in the coming decades, which are estimated at $8 billion (bn) USD to $28 bn USD annually in investment on ships in a transition phase towards decarbonisation in 2050 and about $28 bn USD to $90 USD bn per year for onshore to scale up production, fuel distribution and bunkering infrastructure to supply 100% carbon-neutral fuels by 2050 [25].

There is considerable research on alternative marine fuels' environmental and economic aspects. A study investigated the cost-effectiveness of Net Present Value (NPV) for the top 20 most frequently calling ships to Irish ports in 2019 [26]. The highest NPV is obtained for LNG, followed by methanol and green hydrogen. Green hydrogen is the best option concerning decarbonisation targets, although a further reduction in current fuel price is required to improve its cost-competitiveness over LNG and methanol.

Comparative analysis is also based on in situ data [27]. The data are taken from a Cape-size bulk carrier's operation between Japan and Australia for 30 months. The findings confirm that using LNG as a marine fuel is highly creditable compared with HFO and HFO +Scrubber variants. Results from environmental assessment of alternative marine fuels, including LNG, could be found in [28–31].

As a summary of these studies, the conclusion in [31] can be accepted, which reads, LNG is the main alternative to marine diesel and heavy fuel oil (MDO and HFO) and could provide a cost-effective reduction in $CO_2$ emissions whilst meeting $SO_x$ and $NO_x$ emissions regulations. However, the greenhouse gas (GHG) benefit is reduced by methane slip, with an overall reduction of 8–20% compared to HFO and MDO'. This conclusion can be confirmed by the number of retrofitted and new build ships using LNG as fuel in recent years [32].

The first new-build ship using LNG as fuel was an Offshore supply vessel in 2003 and the first retrofitted ship is an Oil/Chemical Tanker in 2011. Both ships operate in the North Sea.

The life-cycle cost assessment must be considered when evaluating the use of alternative fuels, along with the environmental aspects. Although based on one example of a 1500 TEU container ship, it is concluded in [33] that, in general, LNG-fuelled ships have higher running Operational Expenses (OPEX) and lower Voyage Expenses (VOYEX). A point must be considered the loss of cargo space due to installing LNG tanks onboard.

Recently conducted research [34] provides valuable information for retrofitting a 300 K DWT VLCC trading from the Arabian Gulf to China. The CAPEX as $ USD per tonne deadweight ($/t DW) of competing alternatives is shown in Table 1.

**Table 1.** CAPEX alternatives in retrofit LNG, retrofit open loop scrubber and reference VLSFO.

| Items | Two-Stroke LNG Retrofit | Two-Stroke HFO and Open Loop Scrubber | Two-Stroke VLSFO Conventional |
|---|---|---|---|
| Retrofit CAPEX, $/t DW | 90.67 | 13.33 | 0 |
| Opportunity Cost, $/t DW | 10.33 | 2.33 | 0 |
| CAPEX, $/t DW | 101.00 | 15.66 | 0 |

The opportunity cost is a conservative estimation of the lost charter hire during the 91 day conversion to LNG. The analysis concludes that retrofitting LNG as a marine fuel delivers strong investment returns over the remaining ten years of the VLCC, with returns dependent on relative fuel prices of LNG, HFO and VLSFO. This study introduced the so-called 'Readers' Choice' plot, a drawing for which the reader can select own preferred future fuel price forecasts.

The CII of IMO poses a threat to many VLCC vessels, becoming a stranded asset. Conversion to LNG fuel gives good prospects for these ships. In this case, the LNG alternative maintains a superior rating of B until it slides into C in mid-2027 before finally slipping to D in 2032.

The danger of the vessel overturning in a stranded asset is also analysed in [35]. The total stranded value depends on the size of the LNG fleet and the transition to SZEF. Getting to Zero Coalition plans are for 5% set by 2030. LNG ships must be able to switch to these alternative fuels; otherwise, they will lose their value and the total amount will be more significant if the switch is delayed, for example, in 2034.

The investment decisions about the retrofitting an ageing ship to mitigate the air pollution and transforming the ship into environmentally friendly are to be made based on an informed judgment on the expected economic benefits and the associated risk. Whether the economic benefit is modelled as net present value, return on capital, internal rate of return or economic added value, the risk of a future project is usually represented by a discount rate that reflects the time value of money, i.e., opportunity cost.

Traditionally, the ship is designed based on a selected economic measure of the transported effectiveness using the Required Freight Rate (RFR) established over the years, as seen in [36–39] accounting for direct and indirect costs. The quality of the designed ship is measured by the minimal value of RFR [37], determining the economic efficiency of the investment project which the ship owner has to get to arrive at the assumed profitability rate of a given investment and operating costs, for the assumed ship service life, where the inflation rate, tax rate and the discounted financial balance accounting for the net present value (NPV) and the capital recovery rate are considered.

The use of NPV as a part of the decision support methods is a widespread practice in engineering design which may be confirmed by the review of the state of the arts in [40] and in a specific study in [41–45], among many others.

The feasibility of retrofitting analysis uses the same fundaments as RFR. It is made through the discounted financial balance, based on the net present value, estimated as the sum of expected future cash flows minus the initial investment. The future cash flow is the difference between the expenditure associated with the VLSFO and LNG fuels.

The economic factors involved in the analysis may differ for different geographical locations and economic and political conditions, which is not essential for the method employed here but is needed for the example presented.

An existing ageing multi-purpose ship is used for the present analysis. The recent economic and political instability showed the importance of LNG as an energy source, especially as a marine fuel for the rapidly expanding fleet. The price of LNG as a ship fuel is highly volatile and proper forecasting requires an adequate method. However, last year, the current LNG fuel market showed a severe LNG price increase [46].

The price of LNG as fuel depends linearly on the TTF (Title Transfer Facility: The Dutch natural gas market) [47] and the forecasts of Fitch Ratings for TTF for 2023 are an average of USD$40/Mcf [48]. The price was estimated to be USD$10/Mcf in 2025 from USD$20 in 2024 and stay at USD$5.0/Mcf in 2026 and beyond (1 mt LNG = 48,700 Mcf; Mcf = 1000 cubic feet).

Along with the recent European changes regarding carbon trading in shipping, Japan's proposal at the 78th MEPC session in mid-2022 should also be considered. It applies to a worldwide tax of $CO_2$ of USD$56 per tonne in 2025, USD$135 per tonne in 2030, USD$324 per tonne in 2035 and a shocking USD$673 per tonne in 2040 [49].

The uncertainties originating from the economic conditions are accounted for by employing the extreme value analysis, which can be seen as an approach to assess the cash outflows peaks that can be generated due to unfavourable economic conditions.

## 2. Retrofitting the Propulsion System

The study performed by 'Informal BG' [50] showed that in the Black Sea region, the countries of Bulgaria, Romania, Ukraine, Russia and Georgia reached a container turnover of 3.1 million TEU in 2021, where the Ukrainian annual volume was about 1.0 million TEU. For 2022, the total regional container turnover in the Black Sea is expected to decrease by about 35%. However, the analysis predicted a 10% increase in throughput.

The short sea shipping in the Black Sea region has been analysed in [51]. The study highlighted the continued development of the transport corridors between Asia and Europe. The strategic plan for the year 2030 of the Central Asia Regional Economic Cooperation (CAREC) Program is indicative that the Trans-Caspian International Transport Route, which reaches the Georgian ports of Poti and Batumi, remains of interest to the neighbouring countries, which is an essential link in the multimodal chain.

Improving efficiency and reducing emissions from exhaust gases makes the selected ports of the present study the most representative for the Black Sea region. Additionally, taking into account the recent trends of massive modernisation and construction of new LNG-fuelled ships and despite the speculative rise in LNG fuel prices in recent months, the current trend shows that the UP World LNG Shipping Index—a commodity index for LNG shipping companies—continues to grow (https://seekingalpha.com/article/4559739-lng-shipping-correction-likely-great-sector, accessed on 20 November 2022), confirming that transforming an existing commercial ageing ship to an LNG-fuelled one is very relevant and it will be of extreme priority.

However, some studies about the container traffic and new containership design for the Black Sea region have been performed in [5,10,51–55] and LNG as an alternative for retrofitting ageing ships in [56].

When choosing a ship for retrofitting, the fact of suitable size and age of a ship built near the operating area for this study was initially emphasised, and, in this case, a 9790 DWT multi-purpose ship (Figure 1) equipped for the carriage of containers was chosen given the facts mentioned above.

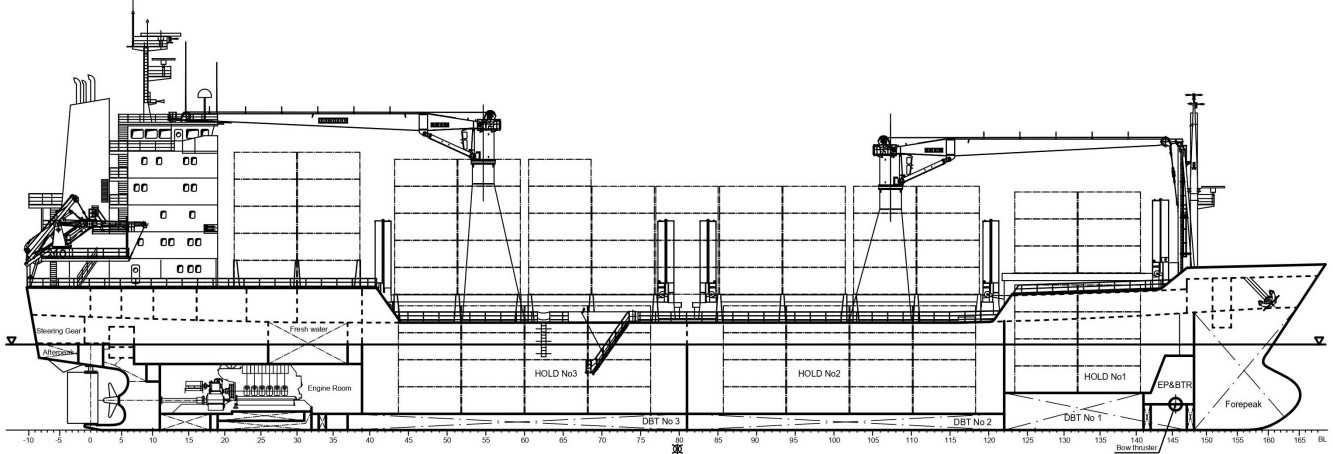

**Figure 1.** Side view of 9790 DWT multi-purpose ships.

The ship was delivered in 2009 and intended for various carriage general, dry bulk, heavy cargo, containers, 40 pcs refrigerating containers on deck, dangerous goods on the main deck and grain. In the study, it is assumed that the navigation area is in the Black Sea.

*2.1. Limitations*

The installation of a power plant with LNG as fuel makes it necessary to adapt the ship's design to a specific set of needs, installing equipment and arranging spaces that would not be necessary in the case of conventional propulsion. At the same time, there are limitations related to the construction and installation of different systems on board. There is also a specific requirement for the ship's safety and people operating retrofitted equipment.

Implementing power plants based on dual engines is technically complex and must comply with demanding requirements regarding their disposal on board, which are imposed by regulation. The versatility of the vessels under study greatly complicates this work since the regular operation of the new equipment cannot interfere in any way with the operations they develop. Limitations are related to a detailed analysis of each system component's location and the study of hazardous areas and escape routes per the applicable regulations.

The preliminary feasibility study addressed in this section has resulted in the integration of LNG technology could become possible for the chosen vessel since it has a larger space and, therefore, offers greater possibilities in terms of generating alternatives for the integration of the necessary equipment for the consumption of LNG on board.

For the installation of the LNG system, the most important fact is to find a place with enough capacity to contain the LNG tank. The selected storage tank is an independent (type C) tank. This type of tank has many possibilities of installation on board due to its portability, being only necessary for a space which fulfils the safety restrictions imposed by the IGF code [57].

Finding sufficient space for storing the gas on board the vessel is a significant factor for the success of the conversion. The LNG storage tank location can be freely selected on board the vessel and either vertical or horizontal tanks, on the open deck or below deck, can be selected. When the storage tank is above deck, the requirements set by the classification societies are slightly lower. However, in our case, it is necessary to install the storage tank above the deck because apart from the structural challenges, the difficulties in choosing the location of the gas storage tanks are also related to the bunkering possibilities at the area of selected routes.

The IMO Regulations [58] were used to locate LNG storage tanks and equipment. According to them, gas storage tanks can be on an open deck or in enclosed spaces. In installing LNG tanks on an open deck, a distance of at least one-fifth of the breadth of the ship from the ship's side and open decks should be used to ensure sufficient natural ventilation and prevent the accumulation of leaking gas in the event of a leak.

Tanks should also be provided with drain pans which should be fitted under the tank and should be of sufficient capacity to contain the volume that may leak out in the event of pipe connection failure.

The bunkering method is essential for determining the location of the LNG tanks. Unfortunately, there is no fully developed bunkering infrastructure in the ship operation area, so only a tank on the deck is considered.

As seen in Figure 1, the ship's superstructure is located as far back as possible so that it does not allow the placement of the gas storage tanks in the stern. On the other hand, the arrangement of such tanks in front of the superstructure is possible, especially since there is an engine room and spaces for auxiliary equipment below the deck, and there will be no requirement to provide space for loading and unloading operations below the main deck. The C-type gas storage tanks are most appropriate for retrofitting the structural arrangement of the multi-purpose ship analysed here.

After the retrofitting, sailing on the route Varna-Poti-Varna is considered and calculated. The maximum fuel consumed for one cycle is 132 m$^3$ of LNG (with included 5% autonomy for bad weather).

The smallest size, i.e., a 30 m$^3$, is selected and the tank is located on the poop deck, in front of the superstructure, taking up space of four containers (TEU). The tank will be isolated from the top side by a specially built metal platform located 4 m from the

deck and designed with a load capacity of 250 tons, sufficient to place four 40ft ISO LNG containers fully loaded and secured by standard foundations and quick-release fittings used on container ships. In addition to the main tank, this permit reaches a capacity of 132 m$^3$ LNG. The volume of the mobile containers is 33 m$^3$ [59].

Such an arrangement meets the requirements, and the length and width of the metal platform are estimated to be 12.2 m and 4.92 m, respectively, while the vertical clearance from the main deck is 4.00 m. The loading and unloading of the adjacent rows of containers are not hindered.

All equipment and systems for regular operation and processing of LNG are installed inside the TCS (tank connection space). It is a closed, gas-tight enclosure, with independent ventilation from the other spaces, made of stainless steel, which acts as a second barrier that prevents a possible LNG leak from affecting the ship's hull.

All LNG storage tanks [59] are highly insulated, but gradual heating is inevitable, leading to BOG (boil-over gas), which must be managed. The BOG can be controlled to a certain point for vessels equipped with Type C tanks by allowing the pressure to build. The IGF Code requires that the tank pressure be maintained below the set point of the pressure relief valves without venting gas into the atmosphere. The IGF code proposes that the minimum holding time for pressure vessels (type C vessels) is 15 days.

Another challenge during the retrofitting project is the selection and positioning of the bunker station. LNG bunker operation used to be a critical situation due to the potential risks this operation involved. Apart from the installation of the station, including all necessary pipes, it is essential to consider that a drip tray must be installed. Drip trays shall be fitted where leakage may occur, which can cause damage to the ship structure or where limitation of the area which is affected by a spill is necessary. The drip tray shall also be thermally insulated from the ship's structure so that the surrounding hull or deck structures are not exposed to unacceptable cooling in case of leakage of liquid fuel. Finally, drip trays must be fitted with a drain valve for rainwater. All these facts are essential to consider when installing on the deck.

A length of 3 m, a width of 1.5 m and a height of 2 m were chosen for the size and dimensions of the bunker station. There are no specific size requirements for these stations and it is sufficient to have room for piping and to provide space for air isolation. The most suitable location is below the deck, where a 1.55 × 1.98 m hydraulic hatch will be constructed to open the station during bunkering. This arrangement of the bunker station will prevent obstructing the deck from passing from the accommodation to the container bays. With this setup, the fuel pipelines will be brought directly to the deck above the bunker station and the construction of additional insulation and casings will not be necessary.

There is a need to install a module related to the use of diesel, such as the module for initial gas ignition, which is also directly related to the gas combustion operation. The module will be located next to the fuel separators in the ER, where the necessary space is 1500 × 800 mm.

### 2.2. Engine Modifications

The ship is equipped with one four-stroke medium-speed, non-reversible type diesel engine with a gas turbine set for supercharging, intended to operate on heavy fuel oil of viscosity 380 CST at 50 °C. The cylinder jackets and covers are cooled with fresh water, and LO cools the pistons.

Analysing the possibilities of retrofitting the ship's power plant, it was found that the model of the ship's main engine—MAK-Caterpillar 6M43C is very suitable for retrofitting into a dual-fuel engine, namely using the new modification of the MAK-Caterpillar 6M46DF, which is essentially the same size as the 6M43C.

The 6M43C has the same footprint as the new M46DF engine. This makes conversion of the existing M43C engines possible and easy to achieve such a conversion holding on to significant components, such as the engine block, crankshaft, air cooler and turbocharger.

These are the remaining parts and the rest are to be renewed for M46DF components. The primary drive is increasing the cylinder bore from 430 mm to 460 mm. Therefore, the following parts will be renewed: cylinder liners, cooling water jackets, pistons, cylinder heads, gas fuel line and engine electronics. Additionally, the following equipment is introduced: big end bearing temperature monitoring, leading bearing temperature monitoring and timing sensors to the camshaft gear wheel and flywheel.

Next to the engine itself, many components must be placed near and next to the engine to make it possible to run the engine on gas. These are:

- The Gas Valve Unit (GVU) controls the pressure of the gaseous fuel towards the engine and ensures safe operation with double block and bleed valves and ventilation possibilities.
- The Ignition Fuel Module (IFM) unit ensures enough filtered fuel oil is delivered to the pilot fuel injection system. The pilot fuel injection system ignites the gaseous fuel.
- Vacuum pump unit. The fuel gas line on the engine and between the GVU outlet and the engine is double walled. This unit creates pressure in the double wall barrier to monitoring any leakage. The extracted air is monitored for gas leakage content and blown off outside.
- Exhaust ventilation module services in the event of an emergency engine shutdown in gas mode. The exhaust pipe after the turbocharger is to be flushed to prevent the accumulation of an explosive mixture in the exhaust pipe.
- Slow turn the device is mounted on the cylinder heads due to the engine's construction, no indicator or over-pressure valves. To detect water on the piston, a slow turn device is mounted to slowly turn the engine before starting.

Additionally, the ship is to be equipped with gas storage tanks, the master gas valve on deck and transfer pumps suitable for LNG, safety devices according to the IGF Code, such as ex-safety zones, double wall gas piping throughout enclosed spaces and inert gas production, storage and deployment equipment.

The comparison of the dimensions of both engines is presented in Table 2, which shows that they are practically the same.

**Table 2.** The primary dimension of both engines.

| Engine Type | Dimensions [mm] | | | | | | | | | Weight [t] |
|---|---|---|---|---|---|---|---|---|---|---|
| | L1 | L2 | L3 | L4 | H1 | H2 | H3 | W1 | W2 | |
| 6M43C | 8251 | 1086 | 1255 | 1583 | 4258 | 1399 | 750 | 2878 | 215 | 94.0 |
| 6M46DF | 8271 | 1086 | 1255 | 1638 | 4258 | 1396 | 750 | 2878 | 215 | 94.0 |

The dual-fuel engine has additional equipment, such as a ventilation module, preignition module, GVU gas supply module, glycol-GU module, BS bunker station, engine slow rotation module, etc. Summing up the weight of the additional equipment, the dual-combustion engine will be heavier by about 4 tonnes, which will not affect the engine's characteristics.

### 2.3. Ship Performance

The operating parameters of both engines are shown in Table 3. The output power of the engines is the same. Additionally, of utmost importance is that the 500 rpm operating speed has not changed, so there will be no need to change propeller shafts, gearboxes and propellers to take power off.

**Table 3.** Operating parameters of 6M43C and 6M46DF engines.

| Performance Data | Unit | 6M43C | 6M46DF |
|---|---|---|---|
| Maximum continuous rating acc. ISO 3046/1 | kW | 5400 | 5400 |
| Speed | 1/min | 500/514 | 500/514 |
| Minimum speed | 1/min | 300 | 300 |
| Brake mean effective pressure | bar | 24.4/23.7 | 21.3/20.7 |
| Charge air pressure | bar | 3.65 | 3.55 |
| Firing pressure | bar | 208 | 150 |
| Combustion air demand (ta = 20 °C) | $m^3/h$ | 33,100 | 32,050 |
| Specific fuel oil consumption | | | |
| $n$ = 100% | g/kWh | 176 | 186 |
| 85% | g/kWh | 175 | 185 |
| 75% | g/kWh | 177 | 187 |
| 50% | g/kWh | 184 | 192 |
| Lube oil consumption | g/kWh | 0.6 | 0.6 |
| $NO_x$ emission | g/kWh | 10 | 10 |
| Turbocharger type | | ABB TPL71 | ABB TPL71 |
| Fuel | | | |
| Engine-driven booster pump | $m^3/h/bar$ | — | — |
| Stand-by booster pump | $m^3/h/bar$ | 4.2/10 | 4.2/10 |
| Mesh size MDO fine filter | mm | 0.025 | 0.025 |
| Mesh size HFO automatic filter | mm | 0.01 | 0.01 |
| Mesh size HFO fine filter | mm | 0.034 | 0.034 |

The specific fuel consumptions are indicated in Table 4, in which the values are given separately depending on the mode of operation of diesel or gas and for the different working loads of the engine, being converted into tons per hour for easy comparison.

**Table 4.** Fuel consumption of 6M43C and 6M46DF engines.

| Load, % | 6M43C | | 6M46DF | | |
|---|---|---|---|---|---|
| | Diesel (g/kWh) | Diesel (t/h) | Diesel (g/kWh) | Diesel (t/h) | Gas (t/h) |
| 100 | 177 | 0.708 | 188 | 0.752 | 0.619 |
| 85 | 176 | 0.598 | 187 | 0.636 | 0.534 |
| 75 | 177 | 0.531 | 189 | 0.567 | 0.480 |
| 50 | 185 | 0.370 | 195 | 0.390 | 0.335 |

It is important to note that the dual-fuel engine has a higher diesel consumption than the currently installed engine. This is a negative point for the dual fuel, which, when running on diesel, will consume more than the current engine, giving the ship less autonomy. The big difference is mainly because the new engine has a pilot pre-ignition system. The speed of a ship depends on the power generated and one can see from the running characteristics of the considered ship in Table 5.

To obtain the gas consumption in tonnes or cubic meters per hour, it is necessary to know the Lover Calorific Value (LCV) and the fuel gas density. Depending on the composition of the gas, these properties may have different values. However, the variation range is small that the average values of LCV = 49.5 (KJ/g) and density of 0.45 t/m$^3$ can be used. The gas consumption in tonnes or cubic meters per hour is presented in Table 6.

**Table 5.** Performance of the ship.

| Load, % | Power, kW | Specific Consumption | |
| --- | --- | --- | --- |
| | | Diesel, g/kWh | Gas + Pilot, kJ/kWh |
| 100 | 5400 | 186 | 7400 |
| 85 | 4590 | 185 | 7524 |
| 75 | 4050 | 187 | 7457 |
| 70 | 3780 | 188 | 7551 |
| 65 | 3510 | 189 | 7646 |
| 60 | 3240 | 190 | 7740 |
| 50 | 2700 | 192 | 7929 |
| 25 | 1350 | 213 | 9379 |
| 10 | 540 | 265 | |
| $NO_x$-Emission, g/kWh | | 10.3 | 2.6 |

**Table 6.** LNG consumption at different loads.

| Load, % | Gas (t/h) | Gas (m$^3$/h) |
| --- | --- | --- |
| 100 | 0.807 | 1.80 |
| 85 | 0.700 | 1.55 |
| 75 | 0.610 | 1.35 |
| 50 | 0.335 | 0.78 |

The natural gas consumption under standard ISO conditions corresponds to the natural gas in the liquid state at 100%, 85%, 70% and 50% MCR, respectively. To convert natural gas into an equivalent gaseous state, its consumption is multiplied by the ratio between the liquid and gaseous state, which is 1/600, leading to a 75% load of 810 m$^3$/h consumption in a gaseous state and a 50% load of 468 m$^3$/h consumption in a gaseous state.

## 3. Gas Emissions

### 3.1. Voyage Description

The Black Sea region is an important economic area with significant potential for using LNG as an alternative fossil fuel for shipping, reducing exhaust gas emissions and complying with the requirements of the IMO and EU regulations for reducing the carbon footprint from transportation.

Around 35% of natural gas and oils imported to the EU are produced in onshore or offshore facilities in the Black Sea region. The development of LNG facilities in this region will soon have a significant environmental and economic impact.

Short sea shipping routes are ideal candidates for Dual fuel technology since LNG supply is frequently available and there is no need for colossal LNG storage on board. The route selection is based on the cycle's length and selected ports between EU and non-EU countries. The length of the cycle of Varna-Poti-Varna is 1234 nautical miles. The parameters of the route are shown in Table 7.

**Table 7.** Voyage parameters.

| Parameter | Varna Quay-Pilot Station | Sea Passage | Pilot Station-Poti Quay |
| --- | --- | --- | --- |
| Distance, nm | 10.0 | 600.0 | 7.0 |
| Engine load, % | 50% | 75% | 50% |
| Speed, kn | 10.0 | 14.3 | 10.0 |
| Time, h | 1.0 | 42.0 | 42 min |

For stopping and sailing from the ports of Varna and Poti, 30 min is provided separately, which will always be conducted on diesel or heavy fuel and will not affect the analysis.

If it is assumed that the average loading and unloading in the port of Poti and Varna takes 20 h, it is assumed that the ship will have the following cycle of operation:

- 21 h (loading, unloading/manoeuvring in Varna—30 min for mooring and unmooring from the port),
- 1.00 h passage in manoeuvring mode—Varna Canal—at 50% load;
- 42 h passage Varna pilot station—Poti pilot station—a speed of 14.3 kn;
- 0.75 h passage in manoeuvring mode—50% load—Poti pilot station—Poti port;
- 21 h (loading, unloading/manoeuvring in Poti port);
- 0.75 h passage in manoeuvring mode—Poti port—Poti pilot station—a speed of 10 kn;
- 42 h passage Poti pilot station—Varna pilot station—a speed of 14.3 kn;
- 1.00 h passage in manoeuvring mode—Varna Canal—at 50% load—a speed of 10 kn;

In this context, the complete cycle is about 129.5 h and there will be a passage of 85.5 h at sea using LNG as fuel. For one month of 30 days, 5.55 complete cycles will be performed between Varna-Poti-Varna and each cycle is 87.5–84 h at a 75% load and 3.5 h at a 50% LNG load. Therefore, it leads to 485.6 h on LNG or Diesel per month, 466.2 h at a 75% load and 19.42 h at a 50% load.

Within one month, the ship will consume 353 tonnes/month of diesel at 75% and 10.1 tonnes/month of diesel at a 50% load.

Natural gas consumption is estimated at 284.3 tonnes/month at a 75% load and 8.4 tonnes/month of gas at 50%. If the vessel uses only natural gas, it will consume 292.7 tonnes per month and 3512.4 tonnes per year.

For a complete cycle of Varna-Poti-Varna, the ship will need 51.2 tonnes/cycle gas at a 75% load leading to 113.78 $m^3$ and 1.16 tonne/cycle gas at a 50% load of about 2.6 $m^3$

The required amount of gas per cycle will be 52.36 tonnes or 116.38 $m^3$ of gas, respectively. Assuming autonomy of 5%, then 122.2 $m^3$ is required in the Varna-Poti-Varna cycle to be placed in the LNG tank.

A summary of the results for the route is presented in Table 8.

**Table 8.** Summary of the results for the two routes.

| Characteristics | Varna-Poti-Varna |
|---|---|
| Distance in miles for cycle, nm | 1234 |
| Consumption of VLSFO for cycle, tonne | 66.0 |
| Consumption of LNG per cycle, tonne | 53.2 |
| Consumption of LNG per cycle, $m^3$ | 118.2 |
| Consumption of VLSFO per month, tonne | 363.1 |
| Consumption of LNG per month, tonne | 292.7 |
| Consumption of VLSFO per year, tonne | 4357.2 |
| Consumption of LNG per year, tonne | 3512.4 |

### 3.2. Energy Efficiency

The EEOI is used for continuous quantitative monitoring of the ship's energy efficiency. It is calculated according to the methodology developed by IMO in MEPC.1/Circ.684 [60]. The indicator is calculated for each voyage as follows:

$$EEOI = \frac{\sum_j FC_j \times CF_j}{m_{cargo} \times D} \tag{1}$$

where: $j$—fuel type; $FC_j$—a mass of fuel used for the voyage, tonne; $CF_j$—coefficient of conversion of fuel mass into the mass of $CO_2$ emissions for fuel $j$; $m_{cargo}$—the transported cargo, t; $D$—distance burned for transport of cargo, nm. The measurement unit of EEOI is [$tCO_2/(t. \text{ nm})$].

An Average EEOI for a given period or the number of voyages is also determined as follows (see Table 9):

$$Average\ EEOI = \frac{\sum_i \sum_j FC_{ij} \times CF_j}{\sum_i m_{cargo,i} \times D_i} \tag{2}$$

**Table 9.** Summary for EEOI for the considered scenarios.

| Characteristics | Route Varna-Poti-Varna | | |
|---|---|---|---|
| Fuel type | VLSFO | | LNG |
| CF | 3.151 | | 2.750 |
| Cargo, tonne | | 7850 | |
| Distance, nm | | 1234 | |
| EEOI | $2.12 \times 10^{-5}$ | | $1.51 \times 10^{-5}$ |

*3.3. $CO_2$ Emissions*

The European Union presented its "*Fit for 55*" package of energy and climate laws, aimed at reducing carbon emissions by 55% compared to the levels from 1990 to 2030 and achieving carbon neutrality by 2050. To achieve the latest EU target, transport emissions need to be reduced by 90%.

This package of directives includes the introduction of the Fuel EU Maritime (FEM) Initiative and the Carbon Border Adjustment Mechanism (CBAM) and amendments to the Renewable Energy Directive (RED) and the Energy Efficiency Directive (EED). Being part of this "Fit for 55" policy package, the European Commission announced the update of Directive 2003/87/EU, which is based on the EU's MRV system [52] (monitoring, reporting and verification of carbon dioxide emissions from maritime transport) and includes the maritime industry in ETS (Emissions Trading System). The EU states that their actions are due to the slow and insufficient progress made by the IMO.

The ETS is a mandatory cap-and-trade system that sets a general limit on greenhouse gas emissions over a specified period.

The Allowances, known as European Union Allowances (EUA), are auctioned between the participants and the number of allowances available compared to the total number of emissions creates a carbon price; the available allowances will decrease linearly about the set emission reduction targets.

The MRV system will continue to apply to ships over 5000 GT regarding the $CO_2$ emissions, being released during voyages to and from an EU port to a non-EU port, to an EU port from a non-EU port, between EU ports and during the ship is moored in an EU port.

Emission factors (*EF*) for marine fuels and LNG will remain constant between MRV and ETS:

- Diesel/gas oil: 3206 $tCO_2$ per tonne of fuel consumed;
- Liquid fuel: 3151 $tCO_2$ per tonne of fuel consumed;
- Heavy fuel: 3141 $tCO_2$ per tonne of fuel consumed;
- LNG: 2750 $tCO_2$ per tonne of fuel consumed.

**4. Economic Analysis**

*4.1. Current LNG Fuel Market*

Simultaneously, with the increase in the number of orders for retrofitting and especially the construction of new ships using LNG as fuel, a severe increase in the price of LNG has been observed in the last year. Figure 2 shows the historical price in USD$/mt for VLSFO and LNG and the difference between them in the port of Rotterdam for the last three years [46].

The recent economic and political instability showed the importance of LNG as an energy source, especially as a marine fuel for the rapidly expanding fleet. The price in the week ending 6 March is 2512.25 USD$/mt (circled in Figure 2). According to the 29 April report from S&P Global Commodity Insights [61], at least the supply balance is expected to support a price above USD$15/MMBtu (ab. 781 $/mt) per year until 2025.

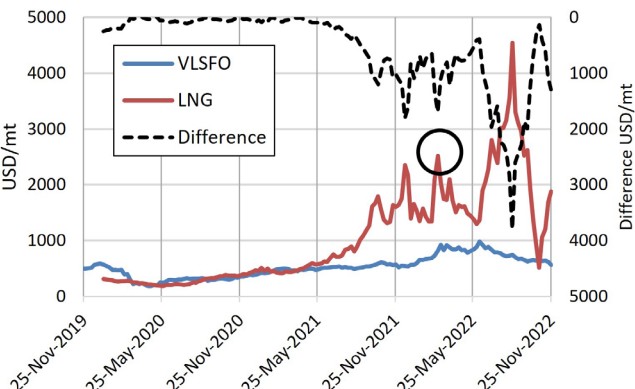

**Figure 2.** Price (USD$/mt) of VLSFO, LNG and the difference (right axis).

The price of LNG as a ship fuel is highly volatile and proper forecasting requires an adequate method. An analysis of possible algorithms is presented in [62].

Standard & Poor claims that LNG is still a viable solution for maritime decarbonisation despite hurdles. They presented in September a report [63] that by 2030, petroleum-based fuels will be 90%, with 7.8% LNG and 2.2% alternative fuels. The optimistic scenario, considering higher $CO_2$ reduction efforts, estimates alternative fuels at 39%, LNG at 32%, LNG at 1% and petroleum-based fuels at 28% of total consumption in 2050. In practice, LNG currently gives the best reduction in $CO_2$ emissions of 23% to 28%.

In the last days of November 2022, there was also a development in the Emissions Trading System. The EU institutions have agreed on including shipping and ETS with expected changes [64]. Ship operators will have to pay taxes in 2025 for 40% of their 2024 emissions, in 2026 for 70% of their 2025 emissions and in 2027 for all their 2026 emissions, with the full coverage will continue after that.

The historical price of $CO_2$ emissions for the last five years are shown in Figure 3. In addition to the price of LNG, there has been significant volatility in prices for $CO_2$ over the last year with a minimum price of 58.341 EUR/mt $CO_2$e in March 2022 and maximum of 98.198 EUR/mt $CO_2$e in August 2022.

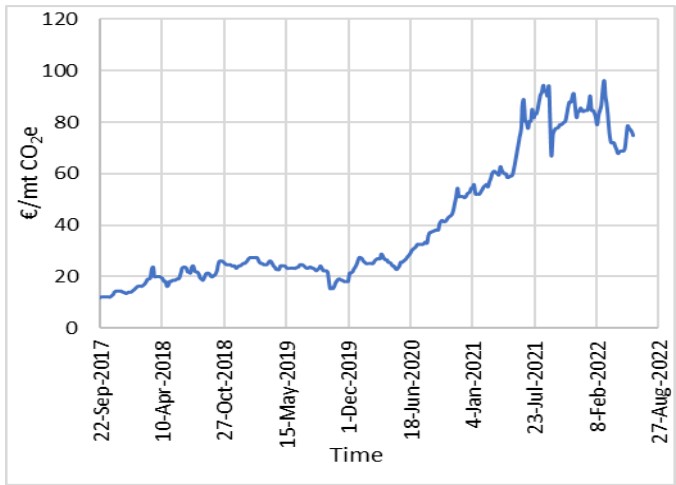

**Figure 3.** Historical price of $CO_2$ emissions (EUR/mt $CO_2$e) for the last 5 years (https://tradingeconomics.com/commodity/carbon (accessed on 20 November 2022)).

Introducing $CO_2$ charges, shipping operators are expected to increase freight rates to cover these costs. If using petroleum-based fuels and 100% emissions coverage, Platts [65] estimates show that, as an example, Aframax freight rates would increase by USD$1.30/mt for the Baltic Sea and the UK and USD$1.80/mt for Black Sea-Mediterranean if the new

ETS rules were in force. Additionally, in this case, it seems natural to take measures to reduce $CO_2$ emissions charges to make the ship competitive.

### 4.2. Feasibility Assessment

The fuel used by ships is priced based on long-term contracts, which are somehow different from the stock market and because of that, four different scenarios will be analysed here.

The cash flow financial approach will be employed to analyse the economic feasibility of retrofitting an ageing multi-purpose ship. Two measures of merit are employed to verify the attractiveness of the LNG fuel compared to the VLSFO, such as NPV and Internal Return Rate (IRR).

The NPV represents the difference between the present value of cash inflows and cash outflows over a period, which was initially formulated by Fisher [66] and used in many studies [67]:

$$NPV = \sum_{t=0}^{n} \frac{R_t}{(1+r)^o} \qquad (3)$$

where $R_t$ is the net cash inflow-outflow, $r$ is the discount rate and $o$ is the number of periods.

It is used to analyse the feasibility of the investment related to the retrofitting here. It also represents the current value of future payments, using the contracted discounted rate and when NPV is positive, the investment is worth undertaking. If it is negative, it is not.

The IRR defines the rate of discount, which makes the present value of the sum of annual nominal cash inflows equal to the initial net cash outflow for the investment. In the present study, IRR is compared to evaluate the profitability of retrofitting an ageing multi-purpose ship from VLSFO to LNG for different scenarios of the fuel market.

If the IRR of retrofitting is lower than the cost of capital, then the best course of action may be to reject it [68]. The risk associated with the IRR is acceptable as a function of the cost of capital and the opportunity cost.

The VLSFO and LNG propulsion systems could add value to the ship's financial performance and one will likely be the more logical decision as IRR prescribes. It was also considered that the implementation of the LNG as a fuel needs to cover the investment related to the retrofitting, which will be the difference between the expenses associated with taxes of $CO_2$ and the current price of the VLSFO and LNG fuels.

The feasibility of retrofitting is made through the standard discounted cash flow approach, based on the net present value, estimated as the sum of expected future cash flows minus the initial investment. The future cash flow is the difference between the expenditure associated with the VLSFO and LNG fuels. The capital expenditure (CAPEX) only related to retrofitting is 3,500,000 USD, an own capital of 500,000 USD is invested, the required net profitability rate is $r = 2\%$ and the resting years of ship operation are $o = 15$ years, the depreciation time is eight years and the time of retrofitting is four months. The average annual inflation rate is assumed as $i_{nfl} = 3\%$, income tax rate $t_x = 15\%$ resulting in a capital recovery factor, $C_{rf} = 7.78\%$ [55] calculated as:

$$C_{rf} = \frac{r}{1 - (1+r)^o} \qquad (4)$$

and the capital cost, $C_{rft}$, is defined as:

$$C_{rft} = \frac{r + i_{nfl} + r\,i_{nfl}}{1 - \left(r + i_{nfl} + r\,i_{nfl}\right)^{-o}(1 - t_x)} \qquad (5)$$

Table 10 summarises the input data for the feasibility analysis, where the assumed parameters are necessary for the case of the particular study but not essential for the methods employed. The original service life for the newly built ship was 25 years, but the retrofitting was performed in the 10th year due to the pressure to reduce the GHG.

**Table 10.** Feasibility input data.

| Description | Value |
|---|---|
| Total Cost of Retrofitting USD$ | 3,500,000 |
| Required net profitability rate, % | 2.00% |
| Number of resting years of ship operation, years | 15 |
| Average annual inflation rate, % | 3.00% |
| Income tax rate, % | 15.00% |
| Capital return factor, % | 7.78% |
| Tax correction $C_{RF}$, % | 8.51% |
| Depreciation time, years | 8 |
| Contract assignment | 1 January 2022 |
| Ship delivery | 5 January 2022 |
| Time of retrofitting, months | 4 |
| Days of operation per year, day | 340 |
| Own investment, USD$ | 500,000 |

The discounted annual average cost of the investment, $C^{aaci}$, is defined as:

$$C^{aaci} = CAPEX\ C_{rft} \tag{6}$$

A systematic variation of the cost associated with VLSFO, LNG fuels and $CO_2$ taxes are employed to analyse the feasibility of the retrofitting project. Four scenarios have been analysed A-Denying, B-Disinterested, C-Good and D-Acceptable, where the first one after ending the service life the NPV is −1,000,000 USD, the second one is with NPV = 0 USD, the third is with 30,000,000 USD and the last one with NPV = 6,000,000 USD. Table 11 shows NPV, IRR and Cash-outflow.

**Table 11.** NPV scenarios.

| Scenarios | Description | Cash-Outflow, $/Quarter, at Fifth Year | IRR, % | NPV, $ |
|---|---|---|---|---|
| A | Denying | 193,183 | 2.13 | −1,000,000 |
| B | Disinterested | 224,143 | 2.99 | 0 |
| C | Good | 317,324 | 5.38 | 3,000,000 |
| D | Acceptable | 410,504 | 7.68 | 6,000,000 |

The LNG fuel price overtook the cost of VLSFO, which recorded a lower price per tonne yesterday. LNG has a lower calorific value than the VLSFO and the prices are not based on the mass but on the gas needed to deliver the same amount of energy, then converted to the oil prices. Additionally, VLSFO and LNG need to pay different $CO_2$ taxes because the analysis here is performed. At this stage, accurate fuel price predictions are challenging since a global gas market does not exist and LNG price levels are highly uncertain.

The achieved NPV and IRR are estimated by defining the expenses difference for using VLFSO and LNG, including the $CO_2$ taxes. Table 11 shows that the Denying scenario generates lost monetary value, which the economic performance of the ship must compensate for. The Disinterested scenario presents neither gain nor loss value and the retrofitting adds no monetary value. The Good scenario adds value and the retrofitting added monetary value equal to the initial investment, CAPEX. In the Acceptable scenario, the retrofitting adds monetary value in a reasonable quantity.

The cost of interest, depreciation, cash-inflow and cash-outflow are also analysed in the present study, reflecting the loss of performance due to age, as seen in Figure 4. The cash-inflow is the sum of the depreciation and the interest rate and the profit will be the difference between the cash-out-flow and the cash-inflow.

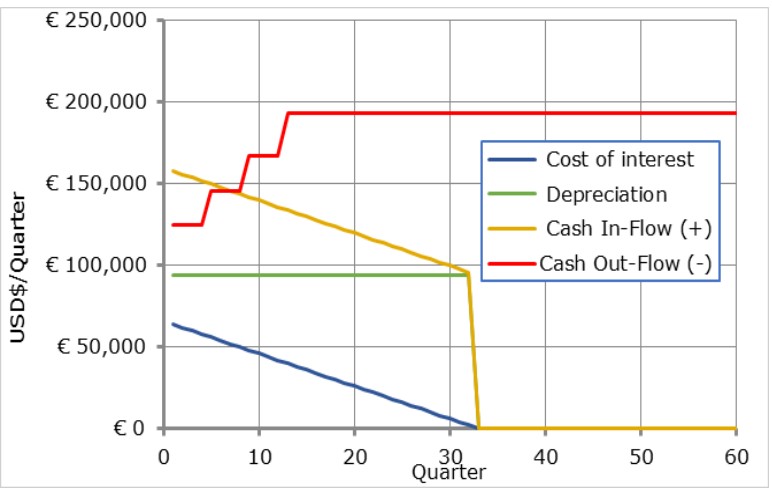

**Figure 4.** Cash-flow, IRR = 7.68%, NPV = 6,000,000 USD$, scenario D.

The cash outflow, $Cash_{outflow}$ is defined as a function of the installed propulsion power, type of fuel and $CO_2$ generated, price of the fuel and taxes related to $CO_2$. For calculating the cash-outflow, factors defining propulsion power, the time of ship operational activity in the voyage duration. The ship analysed here originally used VLSFO and, after retrofitting, used LNG.

The cash-outflow is estimated by multiplying the time spent with the total installed propulsion, engine loading and the operational and consumption for propulsion and summing taxes related to $CO_2$ discounted for the same expenses related to LNG-fuelled engines:

$$Cash_{outflow} = LF_{CO2}C_{CO2}(W_{VLSFO}EF_{VLSFO} - W_{LNG}EF_{LNG}) + (C_{VLSFO}W_{VLSFO} - C_{LNG}W_{LNG}) \tag{7}$$

where $LF_{CO2}$ is load $CO_2$ factor, $C_{CO2}$ is the taxes for $C_{CO2}$ emissions, $W_{VLSFO}$ weight consumption of VLSFO, $EF_{VLSFO}$ is the emission factor for VLSFO, $W_{LNG}$ is the weight consumption of LNG, $EF_{LNG}$ is the emission factor of LNG, $C_{VLSFO}$ is the price of the VLSFO fuel and $C_{LNG}$ is the price of LNG fuel.

The cash-outflow analysis relates the magnitude of a given cash-outflow event with the probability of that event's exceedance. Several assumptions are imposed in analysing the cash-outflow peaks, including data independence, data sufficiency, fuel price description and $CO_2$ taxes for air pollution. The uncertainty propagation in the cash outflow is generated by the stochastic input variables related to the fuel price and $CO_2$ taxes. The cash outflow is estimated for three years, shown in Figure 5, left.

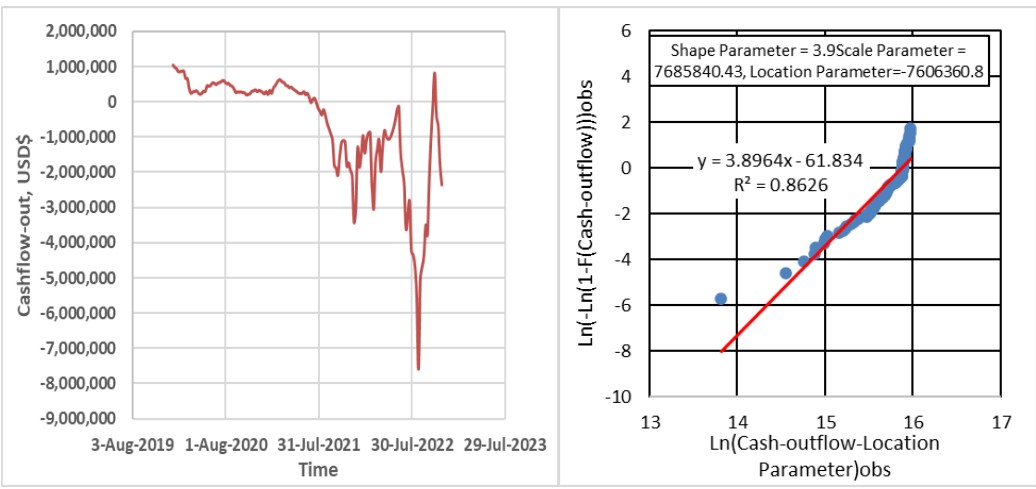

**Figure 5.** Cash-flow peaks (**left**) and Weibull descriptors of cash flow (**right**).

The study uses extreme value analysis to estimate the probability of an unacceptable low cash outflow that can cause serious financial problems. The extreme value analysis can also be seen as a statistical approach to assess the cash outflows peaks that can be generated due to unfavourable economic conditions. In the present study, the Weibull distribution function [45] for describing extreme values of the cash outflow is used due to its simplicity and flexibility. Extreme events may be related to their return period, $T_r$, and for any given return period, the probability of occurrence is estimated as $Q = 1/T_r$. The information on return periods can benefit shipowners in making decisions based on the expected return values. The return values of cash outflow and probability of exceedance are shown in Figure 5. They will be used to predict the cash outflow generated by retrofitting, the probability of exceeding it and the associated financial risk (see Figure 6).

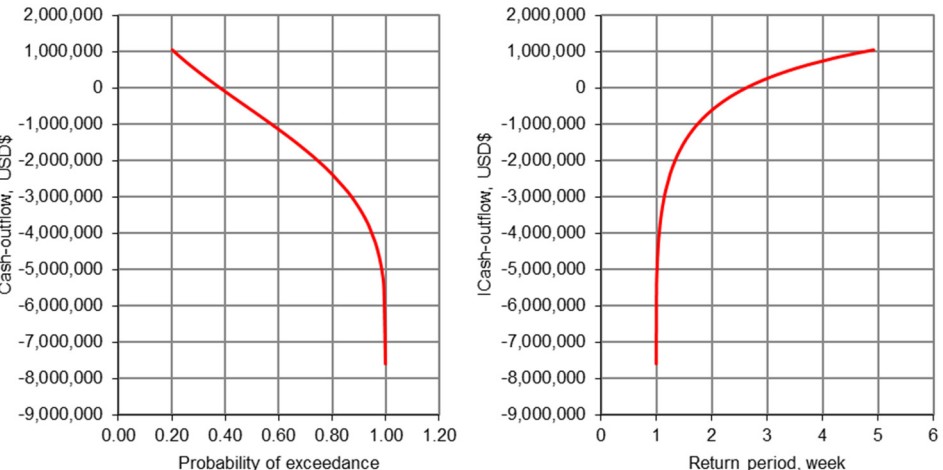

**Figure 6.** Cash-outflow return value (**right**) vs. probability of exceedance (**left**) and return period (**right**).

Additionally, the analysed cash-outflow may be employed to resolve different problems, including defining the minimum cash-outflow needs to be an acceptable limit conditional on the probability of exceedance in a specific period and what will be IRR and NPV to guarantee acceptable benefits of the retrofitting. The analysed scenarios encountered the probability of exceedance and return periods as A (0.39, 2.57 weeks), B (0.38, 2.61 weeks), C (0.37, 2.74 weeks) and D (0.35, 2.87 weeks). It seems the most probable scenarios of the analysed ones is A, followed by B, C and D.

Controlling the cash-outflow level estimates the financial risk of disinteresting in implementing the retrofitting and using the LNG fuel is estimated as follows:

$$R(T_r) = P_r\left(C_{cash-outflow}\,\middle|\,T_r < C_{Lower\ limit}\right)C_{CAPEX} \tag{8}$$

where $R(T_r)$ is the excess lifetime risk in a monetary term for a return period $T_r$, $C_{cash-outflow}$ is the cash outflow, $C_{Lower\ limit}$ = 224,143 USD$ is the lower limit that produces zero NPV and $C_{CAPEX}$ = 3,500,000 USD$ is the consequence of not paying the initial capital investment, which has to be compensated using the ship transportation performance income in monetary terms.

The financial risk for the four analysed scenarios estimated as a function of the cash outflow is defined as A (1,360,219 USD$), B (1,339,762 USD$), C (1,278,634 USD$) and D (121,826 USD$), as can be seen from Figure 7.

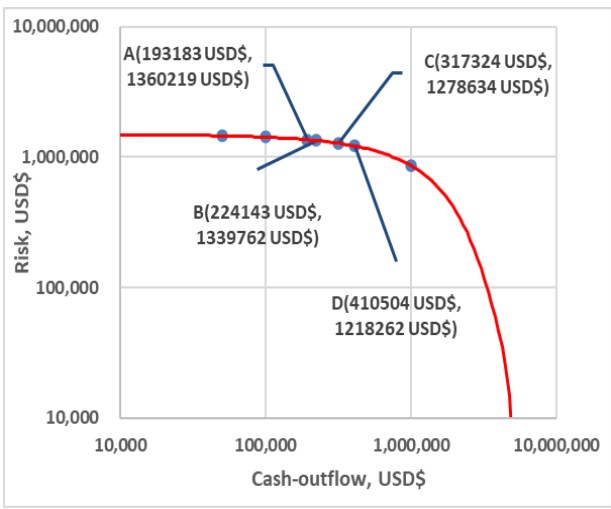

**Figure 7.** Financial risk as a function of cash-outflow.

## 5. Conclusions

The international maritime community, including the leading international institutions, are actively looking to reduce greenhouse gas and carbon dioxide emissions from shipping. Efforts are directed in several directions, such as the design of new ships optimised in size and energy efficiency, innovations in marine engines, utilisation of residual heat, new hull coatings, reducing water resistance and alternative fuels. Optimistic plans aim for 2050 when greenhouse gas emissions are at least 50% of 2008 levels and $CO_2$ emissions will be at 70% of 2008.

The last MEPC (79) started discussion on a revision of the Initial IMO Strategy that is expected to be adopted at MEPC 80 (July 2023). The more important decision on that session is the adoption of amendments to MARPOL Annex VI, acknowledging the whole of the Mediterranean Sea as a designated SOx-ECA (SECA). The amendments will enter into force on 1 May 2024 with the mandatory requirement to use fuel oil with a sulphur content of 0.10% from 1 May 2025.

Recent studies show that the IMO goals can be achieved with alternative fuels that are not petroleum based. However, for the time being, the best alternative fuel in this transition is liquefied natural gas. Despite the high prices of the last year and their variability, the number of refitted and newbuilt ships is constantly growing. Analysts predict that the high prices of liquefied natural gas will not prevent this alternative fuel's introduction into operation.

Along with this, carbon trading in shipping should also be considered. The Council and the European Parliament reached a provisional political agreement on important legislative proposals of the 'Fit for 55' package to include maritime shipping emissions within the scope of the EU ETS and gradually introduce obligations for shipping companies to surrender allowances: 40% for verified emissions from 2024, 70% for 2025 and 100% for 2026.

It is expected that this will also lead to an increase in freight rates and the issue of refitting the ageing ships to improve their competitiveness in the new conditions comes to the forefront.

To answer the question facing every shipowner whether to invest in retrofitting a ship and switching to dual fuel use, a study was made on the economic feasibility of retrofitting a multi-purpose ship operating on short sea routes. After retrofitting with LNG, the economic effect of operating the ship in the Black Sea on the Varna-Poti-Varna line has been analysed, evaluating the four NPV scenarios representing Denying, Disinterested, Good and Acceptable financial cash outflow. It seems that in the present economic instability and price of LNG fuel and $CO_2$ taxes, the ship owner needs to rely on the long-term contract of buying LNG fuel to implement measures to reduce greenhouse gas emissions and keep

good benefits in shipping. The present study introduces a framework that can easily be employed in analysing the economic feasibility of retrofitting ageing ships.

The conducted feasibility analysis and the technical measures related to the retrofitting will be fundamental for the decision-making in implementing the retrofitting and using the LNG fuel and could be used as a risk-based asset management tool regarding the zero-pollution action for the new emission control areas that are planned to be developed in the Black Sea.

However, the methodology used for the feasibility analysis demonstrates limitations in terms of the use of the global and regional information related to the shipping and retrofitting economic and financial factors, which may change suddenly depending on the economic and political conditions and consequently impact the projected trend.

**Author Contributions:** Conceptualisation, D.Y., P.G. and Y.G.; methodology, D.Y., P.G. and Y.G.; validation, D.Y., P.G. and Y.G.; formal analysis, D.Y., P.G. and Y.G.; resources, P.G.; data curation, D.Y.; writing—original draft preparation, D.Y., P.G. and Y.G.; writing—review and editing, P.G. and Y.G. All authors have read and agreed to the published version of the manuscript.

**Funding:** This work was performed within the Research Plan of the Technical University of Varna, funded by the State Budget under the contract NP13 for 2022.

**Institutional Review Board Statement:** Not applicable.

**Informed Consent Statement:** Not applicable.

**Data Availability Statement:** The data presented in this study are available within the article.

**Acknowledgments:** The third author has been supported by the Strategic Research Plan of the Centre for Marine Technology and Ocean Engineering, financed by the Portuguese Foundation for Science and Technology (Fundação para a Ciência e Tecnologia-FCT) under contract UIDB/UIDP/00134/2020.

**Conflicts of Interest:** The authors declare no conflict of interest. The funders had no role in the design of the study, in the collection, analyses, or interpretation of data, in the writing of the manuscript, or in the decision to publish the results.

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
