# Peer review of "Economic Feasibility of Retrofitting an Ageing Ship to Improve the Environmental Footprint"

_applsci, doi:10.3390/app13021199_

Round 1
Reviewer 1 Report
The paper has a very interesting approach to prove, analysing retrofitting LPG to ageing ships is quite good. However, the paper is away from supporting the idea scientifically. The main issue is there is an economic model built on assumptions and the expectation from this model to convince readers that retrofitting LPG to ageing vessels could be feasible. The paper needs a very thorough revision and includes a solid economic implication (not only using the NPV model) to convince the reader that the aim of this paper in fact is scientifically sound.
I have listed some further issues to be improved:
1) Reference to NPV formulation is wrong. This a basic theoretical model initially established by Irvin Fisher, 1907 and many times used by following authors. In this paper, it is attributed to Khan,1993; which is incorrect.
2) Constant and major citation mistakes: in-text citations of web sources are wrong. In an academic paper, this (e.g. line 29-31) is not the correct in-text citation method.
3)Structure of the paper needs a major revision. The introduction includes a literature review, which leads to a low-quality paper structure.
4) Another structural problem is there is no proper sample data introduction. Reader should go through the paper or within lines and find each sample data.
5) Economic Feasibility Section is too assumptive. What's the basis for choosing that r is 2%, operating life 15? Unless that information is provided, the analysis conducted in this paper cannot be classified as scientific.
Author Response
Reviewer 1
The paper has a very interesting approach to prove, analysing retrofitting LPG to ageing ships is quite good. However, the paper is away from supporting the idea scientifically. The main issue is there is an economic model built on assumptions and the expectation from this model to convince readers that retrofitting LPG to ageing vessels could be feasible. The paper needs a very thorough revision and includes a solid economic implication (not only using the NPV model) to convince the reader that the aim of this paper in fact is scientifically sound.
Reply: The investment decisions are made based on an informed judgment on the expected economic benefits and the associated risk. Whether the economic benefit is modelled as net present value, return on capital, internal rate of return or economic added value, the risk of a future project is usually represented by a discount rate that reflects the time value of money, i.e., opportunity cost.
In this context, the present study performs a feasibility study and risk modelling based on the associated revenue from an LNG-fuelled retrofitted ageing ship. The relative risk measure represents the project-specific risk, controlling the cash-outflow level. The outcome of the present study aids decision-making of disinteresting in implementing the retrofitting and using the LNG fuel and could be used as a risk-based asset management tool.
I have listed some further issues to be improved:
1) Reference to NPV formulation is wrong. This a basic theoretical model initially established by Irvin Fisher, 1907 and many times used by following authors. In this paper, it is attributed to Khan,1993; which is incorrect.
Reply: The originally developed formulation about NPV by Fisher [1] has been included in the manuscript.
2) Constant and major citation mistakes: in-text citations of web sources are wrong. In an academic paper, this (e.g. line 29-31) is not the correct in-text citation method.
Reply: The citations were corrected.
3)Structure of the paper needs a major revision. The introduction includes a literature review, which leads to a low-quality paper structure.
Reply: The manuscript's structure has been revised and enhanced, and the literature review has also been improved.
4) Another structural problem is there is no proper sample data introduction. Reader should go through the paper or within lines and find each sample data.
Reply: The description of the sample data is revised, and additional data information is introduced.
5) Economic Feasibility Section is too assumptive. What's the basis for choosing that r is 2%, operating life 15? Unless that information is provided, the analysis conducted in this paper cannot be classified as scientific.
Reply: additional information about the feasibility analysis was introduced. Table 10 summarises the input data for the analysis, where the assumed parameters are necessary for the case of the particular study but not essential for the methods employed. The original service life for the newly built ship was 25 years, but the retrofitting was performed in the 10th year due to the pressure to reduce the GHG.
References
[1] Fisher, I., 1907, The rate of interest, New York: The Macmillan Company.
Reviewer 2 Report
I am sorry that I am not fully in this field. This work seems good to me. Some expression and format should be improved as follows.
1、The GHG in line 22 only has an abbreviation and does not show the full name. The first time it appears, it needs to show the detailed meaning. The same problem occurs in line 67 MEO, HFO, and line 108 DNV.
2、On the contrary, if a word has been clearly explained and given a suggested abbreviation when it first appears, it is not necessary to re-describe it when it appears later. For example, LNG in line 131 and CII in line 158.
3、 The format of the subscript double quotation mark on line 131 and the superscript double quotation mark on line 135 needs to be checked again
4、The left and right figures in Figure 1 are not standardized and confusing.
5、The “a” is missing before "Oil/Chemical Tanker" in line 140.
6、Please unify the table style used for drawing tables.
7、 There are some spelling mistakes in the table. Please correct them. For example, “CO2” in line 429.
Author Response
Reviewer 2
I am sorry that I am not fully in this field. This work seems good to me. Some expression and format should be improved as follows.
1、The GHG in line 22 only has an abbreviation and does not show the full name. The first time it appears, it needs to show the detailed meaning. The same problem occurs in line 67 MEO, HFO, and line 108 DNV.
Reply: The manuscript was verified and, where necessary, was improved.
2、On the contrary, if a word has been clearly explained and given a suggested abbreviation when it first appears, it is not necessary to re-describe it when it appears later. For example, LNG in line 131 and CII in line 158.
Reply: The manuscript was verified and, where necessary, was improved.
3、 The format of the subscript double quotation mark on line 131 and the superscript double quotation mark on line 135 needs to be checked again
Reply: The manuscript was verified and corrected.
4、The left and right figures in Figure 1 are not standardised and confusing.
Reply: The graph was split into two figures for better readings.
5、The “a” is missing before "Oil/Chemical Tanker" in line 140.
Reply: Amended
6、Please unify the table style used for drawing tables.
Reply: Done
7、 There are some spelling mistakes in the table. Please correct them. For example, “CO2” in line 429.
Reply: Amended
Reviewer 3 Report
The GHG emissions of shipping section is a hot topic and has been widely concerned by researchers. This paper showed a demonstration of economic feasibility analysis of retrofitting an ageing ship to a green energy propulsion. The findings interesting, and the conclusion relevant. The topic and findings of the paper are therefore in principle suitable for this journal.
Additional comments:
1. The author/s should highlight the main contributions and originalities of this research at the end of introduction.
2. Some words/sentences in the manuscript are somewhat difficult to understand, including typo, and experienced researchers are suggested to read and proofread them carefully.
3. Please write the full name of the abbreviation when it first appears, and same goes for the whole manuscript.
4. Line 140, was the first ships retrofitted or new build? This is inconsistent, pls check.
5. Lines 180/384, it is recommended to give reasons or citations for this assumption to demonstrate the validity of the method, such as the study area is the ship currently serving, shipping market statistics for the area, etc.
6. At the end of the literature review, the shortcomings of the existing research should be pointed out, leading to the necessity of this study.
7. Section 5, the limitations of this study should be pointed out at the conclusion, far from reducing the value of this work, it increases it substantially.
8. Although this study is analysing a very significant topic in terms of GHG emissions, the discussion of the results should be extended somehow, the practical contributions should be clarified, especially in the Section 5.
The authors had better propose some arguments/contributions associated with management, policy, and/or governance issues relevant to sustainable ocean and coastal development and conservation. Try to give us more inputs about how your interesting information can be used as a management tool.
Author Response
Reviewer 3
The GHG emissions of shipping section is a hot topic and has been widely concerned by researchers. This paper showed a demonstration of economic feasibility analysis of retrofitting an ageing ship to a green energy propulsion. The findings interesting, and the conclusion relevant. The topic and findings of the paper are therefore in principle suitable for this journal.
Additional comments:
- The author/s should highlight the main contributions and originalities of this research at the end of introduction.
Reply: The main contributions are highlighted at the end of the introduction
- Some words/sentences in the manuscript are somewhat difficult to understand, including typo, and experienced researchers are suggested to read and proofread them carefully.
Reply: The manuscript has been carefully verified and amended
- Please write the full name of the abbreviation when it first appears, and same goes for the whole manuscript.
Reply: The manuscript was verified and, where necessary, was improved.
- Line 140, was the first ships retrofitted or new build? This is inconsistent, pls check.
Reply: The text was verified and amended.
- Lines 180/384, it is recommended to give reasons or citations for this assumption to demonstrate the validity of the method, such as the study area is the ship currently serving, shipping market statistics for the area, etc.
Reply: Improving efficiency and reducing emissions from exhaust gases makes the selected ports of the present study the most representative for the Black Sea region. Additionally taking into account the recent trends of massive modernisation and construction of new LNG-fuelled ships and despite the speculative rise in LNG fuel prices in recent months, the current trend shows that the UP World LNG Shipping Index - a commodity index for LNG shipping companies continues to grow[1], confirming that transforming an existing commercial ageing ship to an LNG-fuelled one is very relevant and it will be of extreme priority.
However some studies about the container traffic and new containership design for Black Sea region have been performed by the authors in [1-7] and LNG as an alternative for retrofitting ageing ships in [8].
- At the end of the literature review, the shortcomings of the existing research should be pointed out, leading to the necessity of this study.
Reply: The end of the introduction was enhanced by highlighting the needs of the present study.
- Section 5, the limitations of this study should be pointed out at the conclusion, far from reducing the value of this work, it increases it substantially.
Reply: The limitation of the study has been highlighted in the conclusion
- Although this study is analysing a very significant topic in terms of GHG emissions, the discussion of the results should be extended somehow, the practical contributions should be clarified, especially in the Section 5.
Reply: The practical contributions was highlighted in the conclusion
- The authors had better propose some arguments/contributions associated with management, policy, and/or governance issues relevant to sustainable ocean and coastal development and conservation. Try to give us more inputs about how your interesting information can be used as a management tool.
Reply: The application of the study has been highlighted in conclusion.
References
[1] Damyanliev, T., Georgiev, P., Denev, Y., Naydenov, L., Garbatov, Y. and Atanasova, I., 2021, Short sea shipping and shipbuilding capacity of the East Mediterranean and Black Sea regions. In: Developments in Maritime Technology and Engineering, Vol. 1. C. Guedes Soares and T. A. Santos, editors. UK, London: Taylor and Frances, pp. 749-758.
[2] Garbatov, Y. and Georgiev, P., 2021, Advances in conceptual ship design accounting for the risk of environmental pollution, Annual Journal of Technical University of Varna, 5, (1), pp. 25-41.
[3] Garbatov, Y. and Georgiev, P., 2021, Risk-based conceptual ship design of a bulk carrier accounting for energy efficiency design index (EEDI), Transactions of The Royal Institution of Naval Architects - International Journal of Maritime Engineering, 163, pp. A51-A62.
[4] Georgiev, P. and Garbatov, Y., 2021, Multipurpose vessel fleet for short black sea shipping through multimodal transport corridors, Brodogradnja, 72, (4), pp. 79-101.
[5] Garbatov, Y. and Georgiev, P., 2022, Short sea shipping greenhouse gas emissions and dispersion. In: Trends in Maritime Technology and Engineering, Vol. 2. C. Guedes Soares and T. Santos, editors. UK, London: Taylor and Frances, pp. 35-43.
[6] Georgiev, P., 2022, Development of short sea shipping and multimodal transport of Black Sea region, 17th International Conference on Marine Science and Technologies, Black Sea 2022, Varna, pp. 63-74.
[7] Georgiev, P., Naydenov, L. and Garbatov, Y., 2022, Carbon emissions from container shipping in the Black Sea. In: Sustainable Development and Innovations in Marine Technologies, pp. 85-92.
[8] Yalamov, D., Georgiev, P. and Garbatov, Y., 2022, Liquefied natural gas (LNG) as an alternative for retrofitting ageing ships Proceedings of the 16th International Conference on Marine Sciences and Technologies, Black Sea, Varna Scientific and Technical Unions, pp. 87-95.
[1] https://seekingalpha.com/article/4559739-lng-shipping-correction-likely-great-sector